# Improving Body Representation and Motor Skills with a Preschool Education Program: A Preliminary Study

**DOI:** 10.3390/children9010117

**Published:** 2022-01-17

**Authors:** Ambre Patriau, Juliette Cojan, Thomas Gauduel, Jessica Lopez-Vilain, Gaelle Pavon, Alice Gomez

**Affiliations:** Lyon Neuroscience Research Center (CRNL), INSERM U1028-CNRS UMR 5292, University of Lyon, 69500 Bron, France; ambre.patriau@gmail.com (A.P.); juliette.cojan@gmail.com (J.C.); thomas.gauduel@univ-lyon2.fr (T.G.); jessicalopez.psychomot@yahoo.fr (J.L.-V.); gaelle.pavon@ac-lyon.fr (G.P.)

**Keywords:** body schema, body image, prevention, soft intervention, education

## Abstract

Background: Body representation is described as a fundamental ability to build efficient motor skills. However, no structured and evidence-based program on body representation currently exists. This study assesses the effectiveness of a school-based body representation program (ENCOR: EN for ‘Enfant’ and COR for ‘Corps’ in French) on body representation abilities and motor skills in preschool children. ENCOR focus on body representation abilities as a foundational ability for motor skills. It was designed with teachers and occupational therapists to be autonomously achieved by teachers. Methods: Twenty-three children aged 5–6 years were included and provided with education interventions (control versus ENCOR). Results: Body representation accuracy and precision in localization increased by about 20% and 37%, respectively, in the intervention program compared to the control intervention. In the body part naming task, participants performed fewer of the most frequent errors (i.e., from 198 to 116 left-right discrimination errors). As expected, performance in the body representation tasks and the motor skills tasks were correlated at baseline. We show that motor skills improved after the ENCOR training. Conclusions: Given the need for evidence-based programs in schools, this program could efficiently help implementing body representation education on a large scale. Future studies should evaluate the effectiveness of the program on other cognitive abilities and academic outcomes.

## 1. Introduction

The European commission states that “about 80% of school-age children only practice physical activity and sport in school” [1]. Both international organizations, such as the United Nations Educational Scientific and Cultural Organization, and national organizations, such as the French Ministry of Education, call for the development of students’ motor skills [2,3]. However, many students still retain poor or clearly inefficient motor skills, as in Developmental Coordination Disorder (DCD). More than 5% of children [4] will grow up with poor motor abilities impacting their daily activities and health [5]. In the context of DCD, most solutions have focused on deficits-based interventions rather than the development of educational strategies based on soft universal interventions. Universal programs are economic for a nation, as they guarantee that specialized individualized services (such as psychomotor therapy) are offered, preferably to children with complex needs after resisting 1st-level intervention [6].

This information is even more alarming given the known relationship between motor skills and cognitive development in typically developing children. In typically developing children, sensorimotor abilities have been linked to many critical cognitive functions for school achievement, such as executive functions ([7] but see [8]), mathematical abilities [9,10], time perception [11], auditory processing [12], visual reading processing [13,14], or socioemotional abilities [15]. Fine motor skills in particular have been shown to be essential for cognitive learning [16]. Although arguable, Piaget even proposed that a sensorimotor stage was compulsory for any higher-order knowledge to be acquired [17].

Existing structured programs to improve motor skills, such as the Young Athlete’s Program [18], focus on motor interventions and directly trigger gross motor coordination. They aim to improve balance by coordinating large muscles and movement coordination of the trunk and limbs (e.g., participants perform tasks such as walking, running, jumping, balance, catching, throwing, or kicking).

However, writing with a pen, reading, counting on fingers, navigating in the environment, or whispering a word are all motor actions that require information from a detailed body state representation. Body representation and its integration in our motor skills appear as a foundational ability for motor skills and many curricular activities.

Body representation is a key foundational ability, available at birth [19], that refines during childhood [20]. Our body representation shows short-term plasticity in adults [21,22]. In healthy participants, several studies have shown that body representation and motor skills are linked. For instance, it has been shown that adding a prosthesis with sensory feedback can change body representation and improve motor skills with the prosthesis [23]. In the context of neurodevelopmental disorders, it has been shown that body representation training can induce change in motor skills. For instance, it was shown that using virtual reality to perform movement in an avatar body can induce changes in motor performance in patients with cerebral palsy [24]. In amputees, it was shown that learning movements can, in return, modify body representation of the phantom limb [25]. Recent evidence suggests that body representation in early life has a long-lasting effect on subsequent tool use, such as an artificial arm [26], suggesting that opportunities for sensorimotor plasticity become more limited with age.

Young children exhibit immature behavior with respect to their body representations. For example, until the age of 30 months, children routinely attempt to fit into toy-sized replicas of chairs or cars, suggesting that children at this age do not or incorrectly consider physical dimensions of their own bodies when engaging with the world [27]. However, studies have shown that children can become aware of their own bodies by the end of the second year of their life and that their topographic representation emerges between 20 and 30 months of age [28,29]. Furthermore, it has been shown that by 30 months of age, children can name more body parts (approximately 20 body part names) than the number of body part they can locate on their body. Slaughter and Brownell suggested that this asymmetry would emerge from observing and naming one’s own body parts (in an egocentric perspective) with adult help and guidance to trigger the development of this body part naming knowledge. However, these authors suggest that for body representation to be acquired it would be necessary to reinforce these activities involving a third person perspective to help position and spatially connect body parts to each other [30]. Overall, studies have shown that children’s body representations are not fully developed until the age of 10. Therefore, it may be useful to study how specific early developmental experiences may influence body representation during development.

Moreover, it has been suggested that our motor coordination abilities are intimately related to our body representations. For instance, physical education in school-aged children increases body representation abilities [31]. Effective body representation is an essential component for performing accurate and precise motor programs [32]. Even without motor execution, an efficient and precise body representation is necessary to inform internal models of actions. For instance, it has been suggested that training athletes to refine their body representation may help them to achieve better visualization strategies when creating mental simulations of expert actions [31]. In this context, immature body representation in children may prevent them from performing accurate mental simulation of actions to be performed or to efficiently perform some motor skills (e.g., following verbal instruction to hold a pen such as ‘Your middle finger should grip the pen more lightly than your thumb and index finger”, or imitate movement such as performing a dance in mirror with a sibling).

More recent evidence also suggests that body representation and cognitive skills are directly linked. It was recently shown that integrating an external object as a body part and embedding its functional structure in the motor program can selectively improve syntactic performance in language in adults [33]. It appears essential to promote efficient body representation in children during early development.

Given its importance in motor skills and in early curriculum requirements, various practices to teach body representations have emerged in French schools [34]. However, structured and evidence-based programs of body representation are lacking. To our knowledge no studies investigated the effect of body representation education in young children. We created a body education program (named ‘ENCOR’: EN stands for ‘Enfant’ and COR for ‘CORps’ in French, meaning ‘child’ and ‘body’, respectively). The program is embedded into the French preschool curricula and can be conducted independently by teachers. We aimed to objectively measure the impact of the ENCOR program on body representation and motor skills in order to validate a module of body representation training in preschool children.

## 2. Materials and Methods

### 2.1. Participants

A total of 23 children (5–6 years old) were recruited from a preschool in the Lyon area in France to participate in the study. Children were on average 5.4 years (SD = 0.51; range 5.4–6.3 years). Exclusion criteria were an history of psychiatric or neurological illness, and developmental or learning disorder according to parental reports. They had a normal or corrected vision. Females outnumbered males (13 girls and 10 boys). They participated in the study during the last year of their preschool. On the Edinburgh laterality test (Oldfield, 1971), 15 children were right-handed (7 boys) and 8 were left-handed (3 boys).

The study and experimentation took place within the school during class hours. The study was approved by the Nord Ouest 1 ethic committee (N° 19.12.23.43840). All children and their legal guardians gave informed consent prior to the experiment. The academic inspection and professionals also gave written informed consent to enroll in this experiment. The intervention was run by an experimented teacher (>10 years of experience in preschool) with the help of a territorial agent specialized in nursery schools after several training sessions. The good adherence to the goals of the training delivered was verified by both one of the experimenters (AP) and a district educational advisor.

### 2.2. Design and Procedure

In the within-subject design, children first participated in the body education period (i.e., experimental program) and then, a business-as-usual period (i.e., control program). This design prevents inequality of opportunities between children, as required by the Ethics committee to ensure that all children participated in both programs.

The two programs consisted of (1) four comprehensive lessons (approximately 50 min long, performed twice a week) which includes activities to discover and learn a new notion in depth, and (2) eight routines (approximately 15 min long, a different activity performed every day) to help develop habits, over a period of two weeks. The primary outcome measures were performance on body representation tasks. The secondary outcomes measures were performance on motor skills tasks. All children were assessed at baseline (T1) in March, immediately after the experimental training period (T2) in mid-April and immediately after the control training period (T3), which consisted of a business-as-usual teaching period for the teacher, in the 1st- week of June (See Figure 1). Children were assessed on their ability to name body and hand parts, and to localize on their body after a verbal instruction a body or hand part. Then, visuospatial and fine motor skills were assessed using the NEPSY subtests [35] and Movement Assessment Battery for Children (MABC) subtest [36].

### 2.3. ENCOR Soft Educational Intervention

In the experimental body training program, lessons and routines refer to the foundational skills of body representation. Children were trained to associate naming of body parts with movement and localization of body parts through nursery rhymes (singing the body parts while gesturing [37]), but also to locate body parts by naming, touching, moving them, and locating them on themselves (egocentric perspective) and on one of their siblings (allocentric perspective) to help increase rote association between naming body parts and their location, but also to increase representation of the overall structural organization of their body on themselves and on another body [30]. Children were trained to relate their body size to the size of objects in their environment by estimating whether their bodies could fit into different objects and whether they could pass under different objects while walking, crawling, or walking on their hands and knees [27,28]. Routines aimed to develop tactile precision with or without vision, attendance to proprioceptive afferents (control of body position during relaxation), development of fine movements of body parts (fingers, mouth, faces, muscles, etc.), and development of fine movement of body muscles (contracting and relaxing body parts sequentially). Indeed, the integration of proprioceptive, tactile, and visual information with that from the motor systems is crucial to build an accurate and efficient body representation [38,39].

A detailed version of the body training program is freely available to print for French teachers (https://www.edumoov.com/fiche-de-preparation-sequence/318028/l-oral/ms-gs/developper-le-schema-corporel last accessed: 8 December 2021). It highlights the specific goals, domain, duration, and material for each lesson and each routine, as well as its link with the French educational curriculum. It also states what the instructions are for each part of the lesson, what the professionals (teacher and helper) must do, what material is needed (and provides some of the material needed), what children should do, and what are the potential errors, but also gives some tips.

In the control training program lessons and routines refer to the domain of mathematical abilities.

### 2.4. Measurement

Each child performed a 30 min evaluation at school with one of the experimenters. All children were tested over two school days. Assessment followed the same order for all children: the Edinburgh laterality test, the pointing body part task, the naming body part task, the manual dexterity task (MABC), and the hand posture imitation task (NEPSY).

#### 2.4.1. Body Representation Skills

*Naming Body parts task*: This task is adapted from Sirigu et al. [40]. The child is asked to name as accurately as possible the body part pointed by the experimenter on his body using the tip of a stylet. The experimenter states: “I point with the pen at a part on your body and you tell me what that part of the body is called. If you don’t know, you can still try to answer.”, and give an example: “If I touch you here (left shoulder), what is it?”. Four training trials are proposed to make sure that children understand the task and can ask the experimenter to rephrase if necessary.

In total, 33 items were tested: 23 body parts (nose, knee, etc.) and 10 fingers (left index, left thumb, etc.). For the 23 body parts, first assessed were seven items where no left–right distinction is possible (forehead, stomach, hair, chin, face, neck and nose), then, the child was explicitly instructed to provide a left–right distinction for the other body parts (left elbow, right arm, right thigh, left wrist, left calf, left eyebrow, right forearm, right toe, right shoulder, right foot, left ankle, left hand, right knee, left cheek, right ear, or left leg). The order of body parts is pseudorandomized and the same for all participants. The order was different for each assessment.

The experimenter recorded the name provided by the child. We scored the correct responses. For body parts involving left–right distinction, a point was given only if the child used the correct left–right distinction. The correct response score was computed as the total number of correct responses provided (maximum = 33/33). Errors were categorized as one of the following: (1) a localization error (such as thumb for index), (2) a left-right discrimination error (such left thumb for right thumb), or (3) a superordinate error (such as leg for thigh).

*Touching body parts task*: This task is adapted from Sirigu et al. [40]. The child is asked to point as accurately as possible on his body using his finger the body part named by the experimenter. The experimenter’s states: “I’m going to tell you a body part and you will show me point with your finger this part on your body as accurately as possible. If you don’t know, you can still try to answer”. Example: “Show me your left foot”. Four training trials are proposed to make sure that children understand the task and can ask the experimenter to rephrase if necessary.

In total, 33 items were tested: 23 body parts (eyebrow, nose, knee…) and 10 fingers (index, thumb…). For the 23 body parts, first assessed were 7 items where no left–right distinction is possible (face, belly, nose, etc.), then, the child was explicitly instructed to provide a left–right distinction for the other body parts (hand, arm, foot, etc.). The order of body parts is pseudorandomized and the same for all participants. The order was different for each assessment.

The experimenter recorded on a body or hand diagram the location pointed by the child on his body. The expected answer must fall within what we define as the correct response area, that is the area that can be named by the body parts excluding the intersection with another body part. For each body part, we also defined a central correct response, which is at the center of the body part area. We scored the correct responses as the number of responses within the correct response area. For body parts involving left–right distinction, a point was given only when the child pointed according to his left or right. The correct response score was computed as the total number of correct responses provided (maximum = 33/33). The distance error was computed as the absolute distance (in cm) on the diagram between the location pointed by the child and the expected central correct response.

#### 2.4.2. Motor Skills

*Hand position imitation (NEPSY)* [35]: In this task, children were asked to reproduce a hand posture as accurately and quickly as possible. A maximum of twelve hand postures was proposed for each hand. This requires imitation abilities, digital dexterity, finger recognition, and dissociation abilities, as well as visuo-spatial abilities. A total correct response score (max = 24) was obtained.

*Manual Dexterity subtests (MABC)* [36]: The Movement Assessment Battery for Children assesses motor difficulties in children. The subtest selected from the 4–6 years old range of subtests is the manual dexterity task which requires children to place 12 tokens as quickly as possible in the money bank, using only one hand and taking only one token at a time. We measured the total time to perform the task.

### 2.5. Statistical Analyses

Values are given in mean and standard errors (SEMs). Normality of distribution was tested using the Shapiro–Wilk test. Chi-square (χ^2^) tests were used to assess if some error types were more frequent at baseline. We performed Pearson correlations between body representation and motor skills scores at baseline (T1) and after the body representation training (T2).

To assess the changes in the body representation tasks, we performed an analysis of variance (ANOVA) with a Greenhouse–Geisser correction on the correct responses score with Time (T1, T2, and T3) as a within-subjects factor. If data were not normal, a nonparametric Friedman test was performed. Bonferroni corrections were performed to compare change between T1 and T2 (ENCOR program) and between T2 and T3 (control period). An ANOVA was also performed on the distance error in the touching body part task with time (T1, T2, and T3) as a within-subject factor. To assess whether the distribution of errors changed across time in the body and naming task, we performed χ^2^ tests.

To assess the changes in motor skills, we performed ANOVA with a Greenhouse–Geisser correction on the manual dexterity and hand imitation scores with Time (T1, T2, and T3) as a within-subjects factor. To compare change between T1 and T2 (ENCOR program), and T2 and T3 (control period), we performed Wilcoxon tests with Bonferroni corrections. We estimated effect sizes using η^2^. In all tests, a significance level α < 0.05, after correction was selected.

## 3. Results

### 3.1. Body Representation at Baseline (T1)

At baseline, children correctly named 60% of body parts (mean number of correct responses = 19.70; SEM = 1.9), and correctly localized 54% of body part by touch (mean number of correct responses = 17.7; SEM = 1.28). In these tasks, children performed more left–right discrimination errors (name: 71%, χ^2^(2,3) = 36.72, *p* < 0.001, touch: 59%, χ^2^(2, 3) = 109.73, *p* < 0.001, Table 1) than localization (name: 26%; touch: 26%) or superordinate errors (name: 3%; touch: 15%).

The number of correct responses was positively correlated across body representation tasks (r = 0.897, *p* < 0.001, See Figure 2). Body naming and body touching correct response were positively correlated with manual dexterity (name: r = −0.575, *p* = 0.005; touch: r = 0.573, *p* = 0.005), but not with hand posture imitation (*p* < 0.05).

### 3.2. Body Representation across Time Points

Number of correct responses: A significant effect of Time period was observed for the touch body representation on number of correct responses scores (Figure 3A, Touch: *X*^2^_F_(2) = 10.33, *p* < 0.01; η^2^ = 0.049), but not for the name body representation task (name: *X*^2^_F_(2) = 2.55, *p* = 28). Conover signed-rank post hoc tests with a Bonferroni correction between T1 and T2 showed a significant increase from T1 (Mean = 17.70, SE = 1.28) to T2 (Mean = 21.10; SEM = 1.35; T = 3.21, *p* < 0.01) and no differences between T2 and T3 (Mean = 19.13; SEM = 1.27; *p* = 0.54) on the touch task.

Errors: In the touch task, *X*^2^ test showed that, across time points, the type of errors performed followed the same distribution (*X*^2^ (2,23) = 1.07, *p* = 0.54). A significant effect of Time period was observed for the touch body representation on the amplitude of localization errors (See Figure 3B, Touch: *X*^2^_F_(2) = 13.13, *p* < 0.001; η^2^ = 0.072). Conover signed-rank post hoc tests with a Bonferroni correction between T1 and T2 showed a significant increase from T1 (Mean = 2.20, SE = 0.27) to T2 (Mean = 1.39; SEM = 0.29; T = 3.21, *p* < 0.01) and no differences between T2 and T3 (Mean = 1.45; SEM = 0.28; *p =* 0.558) on the touch task.

In the naming task, *X*^2^ test showed that, across time points, the distribution of error types changed (*X*^2^ (2,23) = 14.50, *p* < 0.01): post hoc showed that the change in distribution occurred from T1 to T2 (*X*^2^ (1,23) = 14.13, *p* < 0.001), but not between T2 and T3 (*X*^2^ (1,23) = 1.24, *p* = 0.53). The percentage of left–right discrimination errors decreased from 71% at T1 to 57% at T2 (Table 2).

### 3.3. Motor Skills across Time Points

A significant effect of Time period was observed for the number of correct responses in the imitation of posture task (*F* (2, 42) = 4.63, *p* < 0.05; eta^2^ = 0.05). Post hoc tests with a Bonferroni correction between T1 and T2 showed a marginally significant increase from T1 (Mean = 11.04, SE = 0.37) to T2 (Mean = 11.90; SEM = 0.42; T = −2.42, *p* = 0.073) and no differences between T2 and T3 (Mean = 12.09; SEM = 0.47; *p* = 1). However, post hoc tests showed a significant increase between T1 and T3 (T = −2.75, *p* = 0.036). We observed no significant effect of Time period on the manual dexterity score (*F* (2, 42) = 1.66, *p* = 0.20).

## 4. Discussion

The study aimed to assess the effectiveness of a school-based body representation program conducted autonomously by teacher on children motor representation abilities and motor skills. Overall, we observed that ENCOR has beneficial effects and induced changes on body representation and improved motor abilities.

Children’s body representation in the touching task improved by about 20% and the precision in localization increased by 37% after only 2 weeks of training. The training did not significantly improve naming accuracy; however, it changed the distribution of errors by reducing left–right discrimination errors in naming. Finally, we observed a near-transfer effect as children increased their imitation motor skills abilities after training with a significant effect between T1 and T3. Body representation abilities have been associated with change in motor skills, and indeed we also found that at baseline, body representation abilities were correlated with motor skills abilities assessed by the manual dexterity score. These results are consistent with models suggesting that body representation is a critical parameter for motor skills [32], and that it is true in our sample of 5–6 years children both before and after an experimental training.

In our study, manual dexterity and naming tasks did not progress after the ENCOR program. If improvement of body representation may have been big enough to be detected through the touching task, it is possible that the change in the naming task was too subtle to be detected by our task in this small sample size. This interpretation is supported by the fact that the error distribution in the naming task did evolve across time points (from T1 to T2 only). As such, qualitative change was observable. The manual dexterity score was selected from the MABC subtests which is well-designed to catch strong deficit in motor performance, but which may fail to capture subtle changes within normal performance. Alternatively, of course, it may well be that body naming as well as pure manual dexterity, per se, were genuinely unchanged despite the improvement in the body representation touching task and hand imitation skills.

The present results suggest that children may have undergone some transition in their mastery of the body representation and its use for motor skills. In fact, in the first assessment, a correlation was found between manual dexterity and body representation correctness. After T2, a new correlation appeared between performance in the manual imitation task and the body representation task (while the correlation between manual dexterity and body representation tasks vanished). It is possible that this change in correlation may point to a reorganization in the integration of body representation to other cognitive abilities such as imitation.

The goal of the ENCOR program is to improve children’s body representation to eventually increase their motor skills, therefore, we did not specifically aim to identify which features promote the change in body representation. Future studies may wish to identify which part of the program most effectively contributes to the global effect. Several topics may be identified, such as the increased association between body part naming, actions, sensing, the raised awareness on children’s body size, the reinforcement of sensory and motor body rituals, and the improved parental monitoring of children body representation.

Motor skills impairment and low abilities in body representation are common in children with Developmental coordination disorders. The “Internal Model” (IM) hypothesis [41] proposed that DCD children fail in motor control because of their difficulty in forming an internal model of the spatio-temporal parameters of the action to perform. Internal models need to be updated with the individual’s body representation, which is important to map movements within an egocentric reference frame [42]. Initial evidence suggests that children with DCD may have imprecise body representation. Indeed, they present sensory disturbances (i.e., tactile and proprioceptive) and poor estimation of body size [43,44]. In addition, they frequently display synkinesis (i.e., involuntary movement of one body part when moving another) across several body parts [45]. Therefore, we hope that future studies can assess whether the ENCOR program can result in changes in body representation and motor skills of children with DCD. Furthermore, a proficient body representation may be an interesting protective factor in children at risk of developing a DCD. As such, we suggest that future studies may assess whether nurturing body representation in children at risk for a diagnosis of DCD may be useful in enhancing their cognitive resilience.

This study faces some limitations and further studies of the program may be conducted to confirm the result. One limitation is linked to the use of a within-subjects design. Such design can be considered as increasing the risk that participants improved because of repeated exposure to the tests and failure to control for effects that vary across time-period. Nonetheless, such design also shows advantages over having another group perform a before–after study in parallel. First, the children included in both programs are the same, hence it reduces confounding error due to natural variance between children. Second, because we did not show any improvement during the control period (between T2 and T3), the risk that the significant effects observed can be attributed to spontaneous improvement over time or a testing effect is greatly reduced.

Future studies with larger sample size may assess the long-lasting beneficial effect on motor skills and body representation. Finally, as discussed in the introduction, body representation may well show direct transfer effect on other cognitive skills (e.g., counting abilities, executive functions, letter recognition, etc.) and on academic performance but that was not assessed in the current study.

Because early school interventions are “crucial instruments to promote […] physical, cognitive, social and cultural development” [1], we suggest that preschool body representation trainings, such as ENCOR, should help policymakers prevent the formation of early gaps in body representation skill, especially for children in disadvantaged situations. Indeed, the ENCOR program is both effective in improving body representation and motor skills, but also very inexpensive because it can be run autonomously by teachers.

In France, teachers have very heterogenous knowledge about body representation and practices that can improve body representations [34]. We suggest that enhanced training of preschool teachers on how to train body representation could be an unexplored avenue to reduce early motor inequalities. Practical and detailed activities to be performed by French teachers are freely accessible on the website (https://www.edumoov.com/fiche-de-preparation-sequence/318028/l-oral/ms-gs/developper-le-schema-corporel, accessed on 8 December 2021) and can easily be translated to other languages.

## 5. Conclusions

The body education program ENCOR integrated into school curricula and autonomously led by teachers can improve body representation and motor skills performance. The ENCOR program is freely available to teachers; it is already implemented into preschool’s curriculum to promote body representation improvement. Furthermore, it is a complementary, efficient method to enhance motor skills in a preventing design rather than wait for children to fall below normal motor skills.

## Figures and Tables

**Figure 1 children-09-00117-f001:**
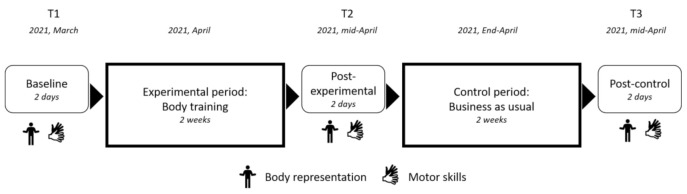
Illustration of the procedure.

**Figure 2 children-09-00117-f002:**
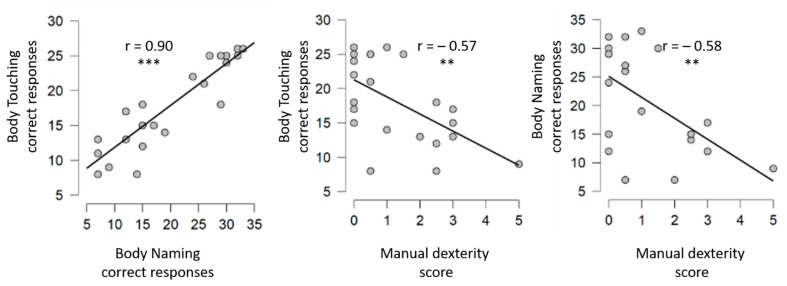
Correlation between body representation tasks and between body representation tasks and motor skills.** is for *p* < 0.01; *** is for *p* < 0.001.

**Figure 3 children-09-00117-f003:**
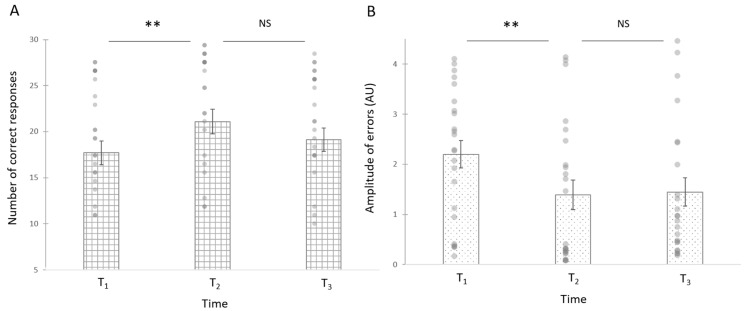
Number of correct responses (**A**) and amplitude of localization errors ((**B**), in arbitrary units, AU) in the touch body part tasks across time period (T_1_ at Baseline, T_2_ After the experimental period, T_3_ after the control period). NS is for non-significant, ** is for *p* < 0.01.

**Table 1 children-09-00117-t001:** Error types at Baseline.

Number of Error and *Percentage (%)* at Baseline	Body Touching	Body Naming
Localization	64 *26%*	72 *26%*
Left-right discrimination	145 *59%*	198 *71%*
Superordinate	37 *15%*	9 *3%*

**Table 2 children-09-00117-t002:** Changes of number of error and percentage (%) with time period (T1, T2, and T3) for each type of error (localization, left–right discrimination, and superordinate). Bold indicate significant change between T1 and T2 on the number of errors.

Number of Error (%)	T1	T2	T3
Localization	72 *26%*	67 *33%*	57 *29%*
Left-right discrimination	**198** *71%*	**116** *57%*	122 *63%*
Superordinate	9 *3%*	20 *10%*	16 *8%*

## Data Availability

Due to ethical agreement with participants and General Data Protection Regulations (GDPR), data are only available for the author’s project and are archived on a local server for 15 years. With the approval from the ethics committee, the data can be shared upon request.

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
