# Peer review of "Improving Body Representation and Motor Skills with a Preschool Education Program: A Preliminary Study"

_children, 2022, doi:10.3390/children9010117_

Round 1

Reviewer 1 Report

ALL : The abstract can be improved. Further studies might provide some applied indications about introduction in some specific aspects. I suggest to review the studies of Raimo S. and Iona T. It's suggested to check the acronyms of some expressions that are not mentioned .

Author Response

  1. The abstract can be improved.

We substantially corrected the abstract as suggested, and we think it significantly improved.

  1. Further studies might provide some applied indications about introduction in some specific aspects. I suggest to review the studies of Raimo S. and Iona T.

We would like to thanks the author for this very interesting suggestion. We added further details concerning body representation development in children and the impact of physical education training on children body representation development as tested by Raimo S. et Iona T.

“Young children exhibit immature behavior with respect to their body representations. For example, until the age of 30 months, children routinely attempt to fit into toy-sized replicas of chairs or cars, suggesting that children at this age do not or incorrectly take into account physical dimensions of their own bodies when engaging with the world world [27]. However, studies have shown that children can become aware of their own bodies by the end of the second year of their life and that their topographic representation emerges between 20 and 30 months of age [28,29]. Furthermore, it has been shown that by 30 months of age, children can name more body parts (approximately 20 body part names) than the number of body part they can locate on their body. Slaughter and Brownell suggested that this asymmetry would emerge from observing and naming one’s own body parts (in an egocentric perspective) with adult help and guidance to trigger the development of this body part naming knowledge. However, these authors suggest that for body representation to be acquired it would be necessary to reinforce these activities involving a third person perspective to help position and spatially connect body parts to each other [30]. Overall, studies have shown that children’s body representations are not fully developed until the age of 10. Therefore, it may be useful to study how specific early developmental experiences may influence body representation during development.

Moreover, it has been suggested that our motor coordination abilities are intimately related to our body representations. For instance, physical education in school-aged children increases body representation abilities [31]. Effective body representation is an essential component for performing accurate and precise motor programs [32]. Even without motor execution, an efficient and precise body representation is necessary to inform internal models of actions. For instance, it has been suggested that training athletes to refine their body representation may help them to achieve better visualization strategies when creating mental simulations of expert actions [31].”

  1. It's suggested to check the acronyms of some expressions that are not mentioned.

We have checked for acronyms that were not mentioned and corrected it.

Reviewer 2 Report

  1. This study deals with the body representation and motor skills for young children. I consider that this issue is very interesting and innovative.
  2. In the section of the introduction, I suggest the author could simplify the policy and statements about the motor skill in school. I suggest the author should provide more explanation about the relationship the body representation and motor skills, and focus on the critical influence of body representation for young children.
  3. In the section of the method, the author took as the experimental method to explore the hypothesis of the related factors. I suggest the author could provide more theoretical statements about the ENCOR, and articulated these theoretical statements as the means of data collecting or analyzing.
  4. In the section of results and discussion, the author found the advantages of ENCOR on the development of motor skills for young children. I suggest the author should provide more practical instructional strategies or policy execution for teachers or preschool policymaker.

Author Response

  1. This study deals with the body representation and motor skills for young children. I consider that this issue is very interesting and innovative.

We would like to thank the reviewer for their interest in this topic and pointing its innovative nature.  

  1. In the section of the introduction, I suggest the author could simplify the policy and statements about the motor skill in school. I suggest the author should provide more explanation about the relationship the body representation and motor skills, and focus on the critical influence of body representation for young children.

The parts on policy statements about the motor skill in school has been reduced.

We also provide more explanation about the relationship between body representation and motor skills:

“Moreover, it has been suggested that our motor coordination abilities are intimately related to our body representations. For instance, physical education in school-aged children increases body representation abilities [31]. Effective body representation is an essential component for performing accurate and precise motor programs [32]. Even without motor execution, an efficient and precise body representation is necessary to inform internal models of actions. For instance, it has been suggested that training athletes to refine their body representation may help them to achieve better visualization strategies when creating mental simulations of expert actions [31].”

We also made a focus on the influence that the immature body representation could have. First by reminding the immaturity of body representation observed in young children:

“Young children exhibit immature behavior with respect to their body representations. For example, until the age of 30 months, children routinely attempt to fit into toy-sized replicas of chairs or cars, suggesting that children at this age do not or incorrectly take into account physical dimensions of their own bodies when engaging with the world world [27]. However, studies have shown that children can become aware of their own bodies by the end of the second year of their life and that their topographic representation emerges between 20 and 30 months of age [28,29]. Furthermore, it has been shown that by 30 months of age, children can name more body parts (approximately 20 body part names) than the number of body part they can locate on their body. Slaughter and Brownell suggested that this asymmetry would emerge from observing and naming one’s own body parts (in an egocentric perspective) with adult help and guidance to trigger the development of this body part naming knowledge. However, these authors suggest that for body representation to be acquired it would be necessary to reinforce these activities involving a third person perspective to help position and spatially connect body parts to each other [30]. Overall, studies have shown that children’s body representations are not fully developed until the age of 10. Therefore, it may be useful to study how specific early developmental experiences may influence body representation during development.”

and then by underlining how this can influence motor skills:

“In this context, immature body representation in children may prevent them from performing accurate mental simulation of actions to be performed or to efficiently perform some motor skills (e.g. following verbal instruction to hold a pen such as ‘Your middle finger should grip the pen more lightly than your thumb and index fin-ger”, or imitate movement such as performing a dance in mirror with a sibling).”

  1. In the section of the method, the author took as the experimental method to explore the hypothesis of the related factors. I suggest the author could provide more theoretical statements about the ENCOR, and articulated these theoretical statements as the means of data collecting or analyzing.

We have increased the theoretical statements underlying the ENCOR program in the method section as proposed by the author and articulated these methodological aspects with theoretical statements from the introduction as follow:

“In the experimental body training program lessons and routines refer to the foundational skills of body representation. Children were trained to associate naming of body parts with movement and localization of body parts through nursery rhymes (singing the body parts while gesturing [37]), but also to locate body parts by naming, touching, moving them, locating them on themselves (egocentric perspective) and on one of their siblings (allocentric perspective) to help increase rote association between naming body parts and their location, but also to increase representation of the overall structural organization of their body on themselves and on another body [30]. Children were trained to relate their body size to the size of objects in their environment by estimating whether their bodies could fit into different object and whether they could pass under different objects while walking, crawling or walking on their hands and knees [27,28]. Routines aimed to develop tactile precision with or without vision, attendance to proprioceptive afferents (control of body position during relaxation), development of fine movements of body parts (fingers, mouth, faces muscles…), development of fine movement of body muscles (contracting and relaxing body parts sequentially). Indeed, the integration of proprioceptive, tactile and visual information with that from the motor systems is crucial to build an accurate and efficient body representation [38,39].”

In the section of results and discussion, the author found the advantages of ENCOR on the development of motor skills for young children. I suggest the author should provide more practical instructional strategies or policy execution for teachers or preschool policymaker.

We would like to thank reviewer 2 for this suggestion. Following this advice we provide further practical instructional strategies for preschool policymakers as follow:

“Because early school interventions are “crucial instruments to promote […] physical, cognitive, social and cultural development”[1], we suggest that preschool body representation trainings, such as ENCOR, should help policymakers prevent the formation of early gaps in body representation skill, especially for children in disadvantaged situations. Indeed, the ENCOR program is both effective in improving body representation and motor skills but also very inexpensive because it can be run autonomously by teachers.

In France, teachers have very heterogenous knowledge about body representation and practices that can improve body representations[46]. We suggest that enhanced training of preschool teachers on how to train body representation could be an unexplored avenue to reduce early motor inequalities. Practical and detailed activities to be performed by French teachers are freely accessible on the website: https://www.edumoov.com/fiche-de-preparation-sequence/318028/l-oral/ms-gs/developper-le-schema-corporel and can easily be translated to other languages.”